# Sorption of Organic Contaminants by Stable Organic Matter Fraction in Soil

**DOI:** 10.3390/molecules28010429

**Published:** 2023-01-03

**Authors:** Aleksandra Ukalska-Jaruga, Romualda Bejger, Bożena Smreczak, Marek Podlasiński

**Affiliations:** 1Department of Soil Science Erosion and Land Protection, Institute of Soil Science and Plant Cultivation–State Research Institute, Czartoryskich 8, 24-100 Puławy, Poland; 2Department of Bioengineering, West Pomeranian University of Technology in Szczecin, Papieża Pawła VI 3, 71-459 Szczecin, Poland; 3Department of Soil Science and Environmental Chemistry, West Pomeranian University of Technology in Szczecin, Słowackiego 17, 71-434 Szczecin, Poland

**Keywords:** atrazine, DDT, chrysene, organic matter, humins, soil contamination, aging, sorption processes

## Abstract

Soil organic matter (SOM) and its heterogeneous nature constitutes the main factor determining the fate and transformation of organic chemicals (OCs). Thus, the aim of thus research was to analyze the influence of the molecular chemodiversity of a stable SOM (S-SOM) on the sorption potential of different groups of OCs (organochloride pesticides—OCPs, and non-chlorinated pesticides—NCPs, polycyclic aromatic hydrocarbons—PAHs). The research was conducted as a batch experiment. For this purpose, a S-SOM was separated from six soils (TOC = 15.0–58.7 gkg^−1^; TN = 1.4–6.6 gkg^−1^, pH in KCl = 6.4–7.4 and WRB taxonomy: fluvisols, luviosols, leptosols) by alkaline urea and dimethylsulphoxide with sulfuric acid. Isolated S-SOM fraction was evaluated by UV–VIS, FT-IR and EEM spectroscopy to describe molecular diversity, which allowed the assessment of its potential sorption properties regarding OCs. In order to directly evaluate the sorption affinity of individual OCs to S-SOM, the mixture of the 3 deuterated contaminants: chrysene (PAHs), 4,4′DDT (OCPs) atrazine (NCPs) were applied. The sorption experiment was carried out according to the 106 OECD Guidelines. The OCs concentration was analyzed by gas chromatography triple mass spectrometry (GC-MS/MS). OCs were characterized by different sorption rates to S-SOM fractions according to the overall trend: atrazine (87.5–99.9%) > 4,4′DDT (64–81.6%) > chrysene (35.2–79.8%). Moreover, atrazine exhibited the highest saturation dynamic with fast bounding time amounting to 6 h of contact with S-SOM. Proportionally, the chrysene showed the slowest binding time achieving an average of 55% sorption after 78 h. Therefore, S-SOM isolated from different soils demonstrated varying binding capacity to OCs (CoV = 21%, 27% and 33% for atrazine, DDT and chrysene, respectively). Results indicate that each sample contains S-SOM with different degrees of transformation and sorption properties that affect the OCs availability in soil. Spectroscopic analyses have shown that the main component of S-SOM are biopolymers at various stages of transformation that contain numerous aromatic–aliphatic groups with mostly hydrophilic substituents.

## 1. Introduction

Increased industrial use and emission of organic chemicals (OCs), along with the increased exposure of environmental and agricultural systems to them, has intensified in recent decades. OCs include many groups of structurally diverse compounds with varying molecular conformation and physico-chemical properties. Some of these compounds exhibit long dissipation half-live times and mutagenic, teratogenic and carcinogenic properties, acting as endocrine disruptors. OCs that are resistant to degradation can cause negative environmental effects due to their susceptibility to dispersion associated with high sorption ability to organic matter [1,2,3]. A direct consequence of their excessive industrial, agricultural or domestic usage causes an increased level of environmental pollution, especiallyin soils [4]. Therefore, the inevitability of soil resources protection is included in the main EU documents ‘Soil Strategy for 2030: Reaping the benefits of healthy soils for people, food, nature and climate’ [5] as well as the assumptions of the European Green Deal [6].

Organic pollutants that draw a lot of attention due to their environmental impact are persistent organic pollutants, which include polycyclic aromatic hydrocarbons (PAHs), organochlorine pesticides (OCPs) and nonchlorinated pesticides (NCPs). PAHs are compounds formed in the incomplete combustion of biomass and fossil fuels and are deposited in soils as a result of natural and human activities (Tang et al. 2022). Compounds from this group, including >4 benzene rings (e.g., chrysene), may cause negative effects to human health and are highly lipophilic. OCPs were used for pest control in crops, the most recognized compounds of this group being 4,4 ‘DDT and its metabolites: 4,4′DDD and 4,4′DDE. Among the non-chlorinated pesticides, the content of atrazine requires control in the environment. It was used as an active substance of many herbicides, being applied for weed control. DDTs and atrazine are currently banned from use in agriculture in the EU, but their residues are still present in the terrestrial ecosystem [7]. All of these contaminants are mainly deposited and accumulated in the soil, where they can remain for a long period of time [2,3,7,8,9]. Studies conducted in different countries showed that, in soils, high concentrations of organic contaminants were found even after many years since their emission or application [3,4,10]. This fact is considered to be relevant in the context of soil health and environmental risk as well as long-term soil management.

Soil organic matter (SOM) and their heterogenous nature constitutes the main factor determining the fate and transformation of contaminants [7,8,9,11,12,13]. SOM is defined as the mixture of organic substances remaining after the advanced decomposition of biomass that includes different organic compounds whose chemical structures and properties depend on in the processes of their formation. Many researchers indicate that OCs–SOM interactions are a limiting factor for the processes of compounds’ bioavailability and degradation [1,2,3,12,13]. Therefore, the persistence of contaminants in soils is mainly determined by the affinity to SOM, which occurs mainly in the humus-rich A horizon. The strength of these processes depends greatly upon many factors, i.e., compound properties (hydrophobicity, solubility, volatility, molecular weight), soil organic matter properties (structural variations, functional group compositions, aromaticity, polycondensation of aromatic rings), and climatic conditions [3]. Processes, such as partitioning, sorption/desorption, sequestration, aging of pollutants, and the formation of bound residue, cause the retention of various groups of OCs in soils at different concentration [11]. Theoretically, OCs may interact with SOM by non-covalent associations (hydrophobic sorption, charge transfer complexes, hydrogen bonding) and the formation of covalent bonds (ester, ether, carbon–carbon bonds) [14,15]. However, the overall process of SOM–OCs complexes formation is often considered a naturally occurring humification of anthropogenic contaminants, since the xenobiotic carbon is embedded in the SOM structure [1]. Thus, the non-soil-originated carbon is sequestered within SOM matrices and is not easy to access by common chemical analytical techniques [3]. With the binding, OCs may also lose their structural identity, including their characteristic physical, chemical and biological properties [12].

SOM may exist in a variety of states, such as dissolved molecules or molecular aggregates, colloidal particles, surface patches or coatings on minerals, intimate complexes with clay-size minerals and discrete particles. At the primary level, SOM is a heterogeneous mixture of functional units within charged, polydisperse molecules that include non-polar alkyl, carbohydrate-like, protein-like, lignin-like, heterocyclic and polyaromatic moieties [12]. These compounds are characterized by different mobility and turnover times in the soil, which makes it possible to distinguish fractions of stable forms (strongly humified, protected by mineral association and occluded within micro and macro aggregates) and labile forms (easily soluble, at the initial stage of transformation). According to literature data [1,2,12], as well as our previous research [16], a stable fraction of organic matter (S-SOM) has the greatest impact on the accumulation and persistence of OCs in soil. Generally, S-SOM constitutes the largest part of total organic matter; nevertheless its chemical characteristics and composition can significantly define the processes of OCs sorption [9]. The type of OCs binding with SOM affects their availability. The OCs sorbed by labile SOM fractions are mainly bioavailable, while the bound and residual fractions of OCs connected with stable organic matter fraction are difficult or impossible to uptake by soil organisms, resulting in a strong limitation of their bioavailability. Thus, knowledge of the mechanisms by which soil pollutants are stabilized by binding to S-SOM is extremely important. The quantities of contaminant-bound residue listed in some literature reviews range from a few percent up to more than 90% of the applied substance [1,3,9]. This indicates that the accumulation of contaminants mostly depends on the mutual relations between S-SOM and OCs, as well as their chemical properties.

The aim of this research was to analyze the influence of S-SOM molecular chemodiversity on the sorption potential to different groups of OCs with regard to their environmental behavior and persistence in soil. The S-SOM fraction was extracted from agricultural soils exposed to a great extent to organic contaminants. Moreover, the S-SOM fraction, as the most resistant part of soil organic matter, was separated from the soil A horizon by sequential extractions with organic solvents, which additionally allowed the exclusion of the influence of the mineral fraction on the studied sorption process.

## 2. Results and Discussion

### 2.1. Spectroscopic Characterization of the S-SOM Fractions

#### 2.1.1. UV–VIS Spectroscopy

So far, there has been little research into the molecular evaluation of a stable fraction of organic matter due to their non-specific character and the need to develop the appropriate separation methods. The obtained spectra (Figure 1) are characterized by the presence of a specific maximum at high intensity within 250–280 nm, which is not convergent with the previous research on the analysis of organic matter components. Thus, it can be assumed that S-SOM is characterized by a specific pattern of spectra with an expressive maximum at low wavelengths. In the higher wave ranges, spectra are broad, featureless and monotonously decrease with increasing measuring range, similar to the graphs obtained for labile organic matter components, i.e., dissolved organic matter, fulvic acids and humic acids [17,18]. Among all samples, two groups with similar spectral properties emerged in the studied spectral range (S1, S2, S3 and S4, S5, S6, respectively) displaying the same basic pattern of wavelength trend.

The main differences between these two groups of samples result from the maximum absorbance intensity, indicating the content of individual groups in S-SOM structures. Generally, the absorption of light by organic matter at low-wave range increases with the increasing degree of the condensation of aromatic rings, the ratio of carbon in the aromatic “nucleus” of the molecule to carbon in aliphatic chains and molecular weight [19]. The efficiency absorbance in the UV–VIS range is mainly related to the chromophore substituent groups at the aromatic benzene ring or aliphatic chains. In organic matter, each aromatic chromophore has three distinct absorption bands in spectra: (1) the local-excitation band at wavelengths < 190 nm, (2) the benzoid band with absorbance between 190 and 230 nm and (3) the electron-transfer band at wavelengths > 240 nm. However, depending on the composition of the chromophore’s structure and type of solvents used, the transition’s energy and molar absorbance will be different [20,21,22]. An increase in the number of aromatic rings and their crowding may results in the batochromic shift of all absorption bands towards the visible range. The presence on the ring of substituents containing lone-pair electrons (OH, OR, NH_2_, NR_2_) includes a batochromic shift with a hyperchromic effect. On the other hand, the alkyl substituents induce a slight shift in the bands toward longer wavelengths [20]. Such behavior may result in the presence of wide bands in the observed samples, confirming the presence of a ‘high congestion of structures’ in the stable organic matter fraction [12].

Moreover, higher absorption in the UV range is related to an increase in the amount of ‘π’ electrons in unsaturated bonds and aromatic fragments of the structure. In addition, acceptor–donor complexes may be responsible for S-SOM in the visible region, which may arise from internal and external molecular aggregation [17,19,23]. These assumptions are reflected in the parameters of the molar absorption coefficients (Table 1). S-SOM exhibited a high value of ε 280 (in range 367–463, 372–563, 366–559, respectively) and a lower value of ε 465, and ε 665 (in range 5–110, 40–76, 24–40, 18–32, respectively). The determined coefficients reflect the high optical density, indicating a greater conjugation of double bonds and the high transformation degree.

#### 2.1.2. FT-IR Spectroscopy

FT-IR spectroscopy allowed the determination of the hydrophobic/hydrophilic properties that indicate the chemical nature of the separated SOM fractions [24,25]. The hydrophobicity is essentially caused by aliphatic C-H groups, present in methyl, methylene and methine units, which are structural elements of alkyl chains, carbohydrates, proteins and lipids [26]. The number of C-H groups in the structure of S-SOM influence their resistance against microbial degradation, rate of wetting, water affinity and the sorption processes of pollutants [24,25,27]. However, the presence of hydrophilic groups also determines the susceptibility of S-SOM to peptization/coagulation processes, which is associated with their resistance to the input of chemical reagents, the mobility in the soil profile and the ability to form chelates with polyvalent cations [26,28].

A higher value of CoV (24.6%) describes the occurrence of hydrophobic components and is indicated by their greater diversity in the S-SOM fraction in comparison to the hydrophilic group characterized by lower CoV (17.0%) (Table 2, Figure 2). The results show that hydrophilic components are predominant, more stable and less affected by changes in soil conditions because they are more susceptible to extraction and hence are distinguished with greater precision. The difference between the value at the maximum and minimum intensities of the hydrophobic compounds of S-SOM was 0.578, while it was 8.817 for hydrophilic components (Table 2). The small contribution of C-H groups in the hydrophobic compounds of the S-SOM structure may be indicated by their limited occurrence in aromatic rings or in their abundance in the functional groups with a high dipole moment. This may also be associated with the presence of additional compounds that stabilize the structure of S-SOM, such as lipids, proteins, plant waxes, pectin, polysaccharides, lignin and other biopolymers [29,30,31]. Hayes [11] claims that S-SOM is mainly simply a less soluble form of other SOM fractions. He also states that, in order to have an understanding of the origins of components of SOM, it is appropriate to consider the composition and structures of the materials and the processes from which these components could be derived and that cause an increase in a wide range of SOM components classified into different fractions.

Generally, a wide range of possible precursors is available in the literature data, but emphasis is mainly placed on the biomolecular structures (cellulose, hemicellulose, lignin, tannins, lipids, cutins and cutans, suberins and suberans, latex materials, algaenan, melanins, char/biochar) that are most likely to contribute to the formation of the stable fractions. It is the view of the authors that, because of their wide distribution, chemical composition and their degree of resistance to microbial degradation, cutins, cutans, suberins, suberans, lipids, algaenans and latex exudates should make significant contributions to insoluble and non-hydrolyzable components in S-SOM [32,33,34,35]. Some or all of these substances are very likely to be major precursors through the selective preservation of aliphatic moieties in S-SOM. Cellulose, hemicellulose, peptides and latex materials can be expected to degrade readily in soils and would need to be protected by sorption or associations with other persistent soil components in order to be found to any significant extent in S-SOM [36,37,38,39]. Altered lignins, tannins and melanins make an inconsiderable contribution to the S-SOM [33]. Char/biochar material, if present, will form part of the S-SOM fraction, but, because it was formed as the result of fire and has not been formed as part of a biological process [40], it is best considered a separate entity though its presence may cause ring forms to be observed in stable SOM fractions.

#### 2.1.3. EEM Spectroscopy

Fluorescence analysis can provide important information on structural and functional similarity and/or differences of individual macromolecules, which can be related to their origin and the degree of transformation [41,42,43,44,45]. Although fluorophore groups constitute only a minor portion of organic matter macromolecules, three-dimensional EEM fluorescence spectroscopy can give spectral information, including the peak intensity, peak location and distribution, as well as information discovered from spectral decomposition and related to electron/photon energy in the fluorescence process [46,47].

The EEM spectra examined the S-SOM fraction divided into two main regions at ex/em of: 280–287/328–344 (peak I) and 316–340/384–412 (peak II), as illustrated in Figure 3, except the S2 sample with one main fluorophore group. Peak I belongs to soluble microbial by-product-like substances, whereas peak II is attributed to humic-like substances, which may indicate their presence as a bound residue in stable forms of organic matter [45,46,47,48,49]. These observations are consistent with the data on the formation of various forms of organic matter, which pointed out that the individual phases that created individual fractions mutually interpenetrate due to chemical dependencies.

Peak I, which is very clearly visible, is associated with biodegradable components characteristic for soluble microbial metabolites and reflects fluorescence from simple structural components of low molecular weight, which can be attributed to amino-acid fluorescence within proteins and lignin or waxes [46]. Peak II refers to non-biodegradable terrestrial organic compounds [45] and can be attributed to the presence of high-molecular-weight components of linearly condensed aromatic ring systems with electron-withdrawing substituents, such as carbonyl and carboxyl groups, and/or to other unsaturated bond systems capable of a great degree of conjugation [46,50]. According to Cory and Mcknight [51], the presence of this band may also be associated with the presence of quinone-like or phenolic fluorophores substituted on both the ring and long aliphatic chain.

The fluorescence intensity for individual samples is presented in Table 3. According to the obtained data, higher intensity of fluorescence is characterized by peak I in the examined S-SOM fractions as compared to peak II. This observation confirms the slight presence of unsaturated bond systems, such as aromatic structures with different types and number of substituents (carboxyl and carbonyl groups), capable of a high degree of conjugation—considered “mature” SOM compounds. Therefore, the proportion of IFl (I) to IFl (II) at the range of 1.17 to 1.78 results from the higher share of low-molecular-weight-component structure with a low degree of transformation and a small number of conjugated chromophores rich in electron-donor substituents (hydroxyl, methoxyl and amino groups) occurring in the analyzed S-SOM fractions [17,46,50].

### 2.2. Sorption of OCs to S-SOM

The efficiency of the sorption process obtained in the experiment for the investigated OCs has been summarized in Table 4. The tested compounds were characterized by a significantly different sorption rate to S-SOM according to the overall trend: atrazine (87.5–99.9%) > 44′DDT (64–81.6%) > chrysene (35.2–79.8%). Atrazine exhibited the highest saturation dynamic with fast bounding time amounting to 6 h of contact with S-SOM (sorption above 98%). Proportionally, the chrysene showed the slowest binding time, achieving an average of 55% sorption after 78 h, while 44′DDT had an average sorption amount at 63.6% after 48 h of contact time.

The sorption equation has reasonably described the adsorption of OCs compounds on the S-SOM, with correlation coefficients (r) ranging from 0.887 to 0.986, 0.876 to 0.989 and 0.902 to 0.924, respectively, for atrazine, 44′DDT and chrysene. The adsorption coefficient expressed in the equation indicates the soil sorption capacity (sorption isotherm slope). A high value of these parameters reflects the high adsorption capacity of S-SOM and thus the lower permeability of soils and the lower leaching potential of contaminants. For chrysene, these parameters were the highest (14.89–23.38), then for 4,4′DDT (9.03–10.82) and the lowest for atrazine (0.067–0.318). These findings indicate that chrysene and 4,4′DDT, despite relatively slow sorption, have a much higher affinity for S-SOM than atrazine. This also means that it is likely that these compounds need a much longer time to diffuse into the S-SOM structures to achieve almost 100% sorption level. So, the observed sorption rate and time is probably controlled by the dimensional structure of the OCs molecule, molecular weight, solubility, vapor pressure and log K_o/w_.

On the other hand, the S-SOM isolated from different soils demonstrated the various binding capacities of OCs expressed by CoV = 21%, 27% and 33% for atrazine, 44′DDT and chrysene, respectively. This indicates that SOM is characterized by structural diversity that affects the retention of contaminants in the soil. Additionally, the observed sorption time dependencies (irrespective of the type of the analyzed compound) were nonlinear and similar to the Freundlich isotherm. According to the Freundlich isotherm model [52], the surface of an adsorbent with a large number of adsorption sites has a wide range of possible adsorption capacity for the contaminant molecules. This will result in the multilayer and heterogeneous adsorption of the contaminant on the adsorbent surface during the adsorption process. The Freundlich model also assumes that the contaminant molecules occupy the stronger binding sites of the adsorbent first, and, as the degree of occupation increases, the binding strength decreases [53]. According to the Freundlich equation superscript less than 1, the adsorption process is considered favorable, showing the formation of stronger interactions between the adsorbent and the contaminant molecules [53], which was observed in our research (Table 4). This is related to the competition for sorption sites on the S-SOM sorbent and the diversified binding strength of contaminants, which, showing high affinity to S-SOM, arrange themselves in adhering layers. Freundlich bond isotherms are compound- and sorbent-specific and describe the strength and nature of the mutual attraction. Nevertheless, the variation in the sorption strength points to the heterogeneous nature of the S-SOM.

### 2.3. Influence of S-SOM Molecular Properties on OCs Sorption Affinity

The relationship between molecular features (measured by UV–VIS, FTIR and EEM) and sorption affinity expressed by maximum sorption amount of OCs to separated S-SOM are shown in Figure 4 and Table 5. All analyzed data were represented by three main components, explaining 93% of the total variance of the results (Table 5), whereas up to 75% of the variance was explained by the first two factors (Figure 4). The first PCA component (PCA 1), which accounted for 45% of the variance, was significantly positively correlated with absorbance indices ε 280/465 (r = 0.96), ε 280/665 (r = 0.67) and ε 465/665 (r = 0.96) and fluorescence parameters IFl (I)/IFl (II) (r = 0.78). Moreover, PCA 1 described the sorption of atrazine and 4,4′ DDT (r = 70 and 71, respectively), which indicates that the sorption of these compounds is influenced by similar physicochemical processes, probably related to their spatial structure equipped by radical substituents (Hofmann et al., 2016) contrary to chrysene.

The second measured component (PCA 2), which represented merely 30% of the OCs sorption variance, was significantly positively correlated with 4,4′ DDT (r = −0.78), as well as the associated parameters of HI (r = 90) and ε 280/665 (r = 0.54), IFl (I)/IFl (II) (r = −0.53). The high correlation of HI and DDT within PCA 2 indicates a strong dependence of DDT sorption on the components of organic matter characterized by the hydrophobic nature derived from numerous short aliphatic chains (lignin, proteins, waxes) obtained in the first stage of the decomposition of plant residues. These relationships may influence the processes of strong sorption and explain the very long time persistence of 4,4′DDT in the soil.

The third PCA component (PCA 3), explaining only the variability of chrysene sorption (r = −0.77), showed a negative correlation with the coefficient ε 280/665 at the level r = −0.71, which indicates that this compound is sorbed in the phases of organic matter with high optical density, indicating a greater conjugation of double bonds and high transformation degree. The previous research of the authors [16] supports this thesis, because chrysene was strongly bound to aromatic fractions of organic matter with short multiple bonds. In this case, the organic matter was isolated through analytical procedures and separated from the mineral forms responsible for its binding and protection. Hence, it may result in weaker chrysene bonding by S-SOM than other OCs.

All of these findings pointed out that the chemodiversity of S-SOM impacts the accumulation of OCs in varying degrees. The heterogeneity of SOM is a property related to its chemodiversity due to the presence of various chemical groups that form sorption domains characterized by different sorption selectivity. This hypothesis holds that strands of SOM associate together on the basis of functional group identity to form domains large enough in scale to act independently as a micropartition phase, e.g., carbohydrate-like domain or aromatic domain. Several researchers identified the structure of SOM as two condensed phases—a hard/glassy/condensed phase and a soft/rubbery/amorphous phase—containing different lengths, densities and reactivity of the aliphatic side chains relative to the soil organic components [12,14,15]. The condensed domains originate from the remnants of plant cuticular waxes, cutan, cutin and subarin, which are long preserved in SOM, especially in the S-SOM fraction. Sorption in condensed domains generally is hindered due to the high rigidity of the structure and its high density. Hence, the slow sorption of chrysene and 44′DDT was characterized by the highest molecular weight among analyzed OCs. Moreover, the condensed phase forms regions of disordered side chains, which have a high density but low reactivity and flexibility due to the presence of numerous unsaturated bonds [11,12,14,15,29], which also restricts the sorption of molecules with low reactivity due to the small number of substituents or their absence (again chrysene and 44′DDT). On the other hand, atrazine undergoes rapid sorption, probably resulting from a lower molecular weight and higher reactivity of the amino and methylene substituents in the structure of this compound (Table 2).

Additionally, the side chains with a high degree of branching may condense with the OCs molecules combining to each other throughout the covalent bonds and thereby reduce the intermolecular interactions by building them into the SOM structures [12,14,15]. The amorphous phase is dominated by carbohydrate-like and lignin-like moieties that are intimately mixed and therefore do not form independent sorption domains. The high content of lignin and aliphatic compounds in the analyzed S-SOM fractions has been confirmed by the UV–VIS, FTIR and EEM techniques, and the results are presented in Section 2.1.1, Section 2.1.2 and Section 2.1.3. It can therefore be assumed that these groups dominate in the studied fractions, strictly influencing the greater preference to sorb the atrazine. Nevertheless, the interpenetration of condensed and amorphous phases cause more active and stable binding sites, which allow for the effective occlusion of other molecules but over a much longer period of time [12,23].

The obtained results confirmed that the properties of the tested contaminants but also the heterogeneous structure of organic matter control the sorption rate. Thus, the analyzed processes were considered in terms of both chemical actions between a compound and a sorbent in which molecules penetrate and are retained by the S-SOM through different chemical forces that determine mutual affinity. Nevertheless, the range of this interaction depend on the organic matter state (dissolved or solid), age (young or old) and other factors [12,14,15] controlled by environmental conditions and soil properties.

## 3. Materials and Methods

### 3.1. Soil Sampling

The six soil samples (S1–S6) were collected from the surface layer (0–30 cm) of agricultural lands, including arable fields. The distribution of sampling points aimed to reflect various soil properties, preliminarily based on maps (1:25,000 scale; database of the Institute of Soil Science and Plant Cultivation). For each sampling site, six replicate samples were collected at the center and vertices of a 1 m × 1 m square, averaged on the site after the removal of the upper layer of organic vegetative materials, and mixed to provide a bulked sample for the site. The collected soil samples were air-dried and passed through a 2 mm mesh sieve. All samples were then stored in the dark at 12–16 °C before the laboratory analysis.

### 3.2. Chemical and Physical Analysis

#### 3.2.1. Soil Physicochemical Analysis

The soils were analyzed for the particle size distribution, pH and organic matter concentration including: determination of total carbon (TC) and total organic carbon (TOC,) and total nitrogen (TN). Thus, the pH was measured potentiometrically in a 1:2.5 (m V^−1^) soil suspension in KCl (PN-ISO10390, 1997). The particle size distribution was analyzed via the aerometric method (PN-R-04032, 1998) to determine the soil A horizon texture; total organic carbon content was determined after sulfochromic oxidation, followed by the titration of the excess of K_2_Cr_2_O_7_ with FeSO_4_(NH_4_)_2_SO_4_·6H_2_O (PN-ISO 14235, 2003). To express their mutual proportion, TN and TC were analyzed by the dry combustion method on a TC/TN Vario Macro Cube analyzer. All measured soil parameters are included in Table 6. The soils were characterized by different physicochemical properties (TOC = 15.0–58.7 gkg^−1^; TN = 1.4–6.6 gkg^−1^, pH in KCl = 6.4–7.4 and WRB taxonomy: fluvisols, luviosols, leptosols), which allow finding potential variations in the characteristics of the secreted stable organic matter fraction.

#### 3.2.2. Extraction Procedure of Stable Organic Matter

In order to reliably assess the studied dependencies, the S-SOM fraction was separated from the soil in order to isolate the most resistant fraction of soil organic matter and, consequently, exclude the sorption potential of the mineral fraction.

The S-SOM was extracted by alkaline urea and dimethylsulphoxide (DMSO) with sulfuric acid, according to the method described by Song et al. [54]. Thus, the dried and grated soil sample (20 g) was first exhaustively extracted, by 0.1 M HCl to pH = 2 and then 0.1 M NaOH-adjusted to pH = 7 to discharge isolate mobile and easily soluble organic matter fractions. Extraction was carried out in several cycles, about 15 times each, until the solutions were completely discolored.

The soil residues were further exhaustively extracted with 0.1 M NaOH + 6 M urea (base urea) using a soil/solution ratio 1:5 (all extractions in basic media were carried out under N_2_). The pH of the isolate was adjusted to 1.5 (6 M HCl) and discarded. According to the methodology, the SOM components exhibit a tendency to concentrate in soil mico- and macro-aggregates. Therefore, the recalcitrant organic fraction was concentrated in the dark clay/silt surface layer after the centrifugation of humic-like substances. This layer was isolated, air-dried, mixed with DMSO + 6% (*v*/*v*) H_2_SO_4_ (98%) (DMSO + H_2_SO_4_; soil/solution ratio 1:10), shaken for 12 h and centrifuged. The process was repeated until the supernatant was discolored. The obtained extract was diluted in distilled water to pH = 2 and then the residue containing the complex of DMSO:S-SOM was isolated by centrifugation, washed with distilled water, de-ashed with 10% HF and freeze-dried.

The obtained cleansed S-SOM fractions were then characterized using spectroscopic methods and subjected to sorption batch experiments with organic contaminants selected for the study.

#### 3.2.3. Characteristics of the Isolated Stable Organic Matter Fraction

The separated S-SOM were characterized by elemental and structural arrangement to evaluate their sorption properties. Ultraviolet–Visible spectroscopy (UV–VIS), Fourier-transform infrared spectroscopy (FTIR), and excitation–emission matrix (EEM) fluorescence spectroscopy were used for this purpose. The application of tese methods enabled the assessment of S-SOM structures’ chemical diversification in the formation of specific moieties of chemical compounds that potentially determine the organic matter sorption processes.

##### The UV–VIS Spectroscopy

The UV–VIS spectra were recorded for the determination of molecular structure diversity in the form of aromaticity and maturity degree of S-SOM fractions. The UV–VIS spectra were measured with the UV–VIS–NIR Jasco V-770 spectrophotometer (Easton, United States). The absorbance measurements were carried out in the range of 230 to 700 nm at a constant concentration of 0.01 mg C·cm^−3^ in a DMSO + 6% (*v*/*v*) H_2_SO_4_ (98% wt) solution in a quartz cuvette with an optical path length of 1 cm. Before performing the analysis, the S-SOM solutions were pre-filtered through a syringe filter with 0.45 µm pore size in order to obtain a high homogeneity of the sample. On the basis of the obtained absorption spectra, the following coefficients: ε280, ε465, ε665 and their mutual proportions were calculated according to the absorbance ratio.

##### The FT-IR Spectroscopy

The FT-IR spectra were recorded for S-SOM due to their specific chemical character and least-known complex nature. Individual spectra were measured in the absorption mode on KBr pellets in the wave number range of 4000 to 400 cm^−1^ using a IR300 FT-IR spectrometer (Thermo Mattson, Madison, WI, USA). The KBr pellets were obtained by pressing, under a reduced pressure, a mixture of 1 mg of freeze-dried S-SOM and 300 mg KBr (spectrometry grade). To minimize the interference from water, the KBr was dried by heating (at 105 °C) and kept under vacuum in the desiccator prior to use. The recording was performed with a resolution of 4 cm^−1^ and 20 scans per sample. Each spectrum was corrected on the ambient air as a background spectrum. The hydrophobicity index (HI) was calculated according to Capriel [26]) and Matějková and Šimon [55] as the ratio of the area of C-H bands that occurred in the range of 2950 to 2830 cm^−1^ to the area of C-O bands that occurred in the range of 1770 to 1560 cm^−1^. The areas of the absorption bands, regarding the hydrophobic (CH-groups) and hydrophilic (CO-groups), were integrated with Omnic, version 6.0, spectrometer software (Thermo Nicolet, Madison, WI, USA) and were defined as intensities.

##### The EEM Spectroscopy

Currently, three-dimensional EEM fluorescence spectroscopy provides a comprehensive and complete view of all the features present in the selected spectral range; therefore, it is often referred to as a “molecular fingerprint” for many different types of samples.

The EEM were recorded on a Hitachi F-7000 fluorescence spectrophotometer (Chiyoda, Tokyo, Japan). The EEM spectra were scanned at emission wavelengths from 250 to 600 nm (10 nm interval) by varying the excitation wavelengths from 250 to 600 nm (10 nm interval). The spectra were recorded at a scan speed of 1200 nm min^−1^, and the scanning interval for excitation and emission was 10 nm. The spectral recordings were performed at a constant concentration of 0.01 mg C·cm^−3^ in a DMSO + 6% (*v*/*v*) H_2_SO_4_ (98%wt) solution in a non-fluorescent quartz cuvette with an optical path length of 1 cm at room temperature. Before performing the analysis, S-SOM solutions were pre-filtered through a syringe filter with 0.45 µm pore size. The EEM spectra were processed to a higher resolution in QUANTUM GIS program v. 3.10. The TIN interpolation was used in order to increase the resolution spectra. The SAGA GIS program (Hamburg and Göttingen, Germany) was used for smoothing the spectra distortions on individual raster.

### 3.3. Experimental Design

#### 3.3.1. Tested Compounds

In order to assess the sorption affinity of individual OCs to S-SOM, the mixture of the three deuterated organic contaminants: chrysene from the PAHs group, 4,4′DDT from the OCPs group and atrazine from the NCPs group was applied. The selected compounds are widespread in the environment due to their historical usage in agriculture. Moreover, they represent different groups of contaminants with specific properties reflecting the behavior of these compounds in the natural soil ecosystem, e.g.,: octanol/water partition coefficient indicating the sorption potential in the organic phase of the soil; half-life time determining the susceptibility to degradation and the rate of this process; and hydrophobicity influencing the ability of the compound to migrate within the soil profile, as well as the ability to bind with organic components. The chemical characterization of tested compounds is summarized in Table 7.

#### 3.3.2. Method for OCs Determination

Determinations of OCs compounds were carried out using gas chromatography triple mass spectrometry (GC-MS/MS; Agilent 7890 B GC system; Agilent Tech., Santa Clara, CA, United States), equipped with an Agilent 7000 C detector and Agilent 7693 Autosampler (Santa Clara, California, United States). Table 8 displays the GC, backflush and MS/MS method parameters. GC was configured with a multimode inlet (MMI) equipped with an 4 mm ultra-inert liner, splitless single taper and a glass wool liner (p/n 5190–2293). From the inlet, 2 HP-5 ms UI columns (0.7 m × 150 μm, p/n 160–2625-5; and 30 m × 250 μm × 0.25 μm, p/n 19091S-431 UI, Agilent Technologies) were coupled to each other through a purged ultimate union for the use of mid-column/post run backflushing. The identification of atrazine, chrysene and 4,4′DDT compounds were performed in multiple reactor monitoring (MRM) mode with individual diagnostic ions. Samples were analyzed up to 24 h after their recovery in order to reduce the possibility of the decomposition of contaminants. To ensure the quality of the compounds identification, the PCB 155 compound was used (2,2′,4,4′,6,6′-hexachlorobiphenyl, Dr. Ehrenstorfer GmbH, Augsburg, Germany) as a surrogate standard (samples fortified at 5 ug ml^−1^ directly before sample concentration) to estimate and control the matrix effects in the analyzed samples. Additionally, PCB 207 (2,2′,3,3′,4,4′,5,6,6′-nonachlorobiphenyl, Dr. Ehrenstorfer GmbH, Augsburg, Germany) was used as an internal standard (added in the same concentration to the sample extract directly before injection) that allowed the control of the instrument and injection GC MS/MS parameters.

The precision of the method expressed as a relative standard deviation (RSD) was in the range of 2 to 5%, and the recovery for individual compounds for certified reference solution was within 90–95%. The limit of detection (LoD) for individual OCs compounds was at the 0.01 µg kg^−1^ level. The detection limit value was adopted as the minimal value of the content of measured OCs compounds.

#### 3.3.3. Sorption Experiment

The sorption experiment was carried out according to the Guidelines for the Testing of Chemicals OECD no. 106 over 0, 6, 12, 24, 48, 72, 78 h in darkness and constant temperature conditions (20 ± 1 °C). The mixture of the OCs was dissolved in hexane (test solution) and added to S-SOM isolated fractions at a concentration level of 10 µg·mL^−1^ and 1:1 volume ratio, e.g., 20 mL of OCs solution and 20 mL S-SOM fraction. The used mixture was a system of two immiscible liquids due to the different polarity of the solutions, at the same time ensuring the free diffusion of compounds at the liquid–liquid interface striving to achieve a state of dynamic equilibrium through the sorption of OCs to S-SOM.

The parameters of the sorption processes were evaluated based on the optimization experiment with the assumption that the concentration of the test compound in the solution should not exceed half of its solubility and be at least two times higher than the limit of detection of the GC MS/MS. The quality control parameters of the experiment were ensured by the inclusion of blank samples, such as reagent and matrix blanks, as well as performing the experiment in three replications for each individual mixture.

The quantitative assessment of the content of contaminants adsorbed on the studied fractions of S-SOM was determined in the test OCs-spiked solution by GC MS/MS after each time period. The adsorbed concentration of OCs was calculated from the difference between the initial concentration in the test solution and the concentration after a specified time. The initial concentration was considered to be the amount of OCs present in the control (without S-SOM).

### 3.4. Statistical Analysis

The software package Statistica (Dell Statistica, version 13.3) was used for statistical analysis. Basic statistical parameters, such as mean, median, lower quartile (LQ), and upper quartile (UQ), standard deviation (SD) and coefficient of variation (CoV), were calculated. Spearman’s correlation was used to assess the strength of the dependence of OCs to S-SOM sorption over time described by an exponential function according to the assumption of the Freundlich equation (Venkanna and Swati, 2010). The one-way analysis of variance (ANOVA) with the U Mann–Whitney test was used to evaluate the difference between the sorption affinity of individual OCs. Moreover, principal component analysis (PCA) was applied to assess the relationship between the molecular sorption properties of isolated S-SOM and the sorption affinity expressed by the maximum sorption amount of OCs.

## 4. Conclusions

The obtained results indicate that each of the analyzed soil samples contains organic matter characterized by various degrees of transformation and chemical properties and thus may interact with organic pollutants in different ways and determine their availability in the soil. Spectroscopic analyses have shown that the main component of S-SOM are biopolymers at various stages of transformation, which contain numerous aromatic–aliphatic groups of substituents that are mostly hydrophilic in nature. These properties significantly influence the sorption behavior of OCs and thus their potential to accumulate in soils. The sorption rate expressed by the concentration of binding compound and the time of this process differs depending on the properties of OCs (the presence of substituents, thus the more effective sorption of atrazine and 4,4′DDT) and the characteristics of S-SOM variability resulting from the properties of the soils from which they were isolated. Atrazine was characterized by the fastest sorption in contrast to 4,4′DDT and chrysene, but the affinity of individual compounds to S-SOM resulting from the sorption equation was inversed. These specific relationships significantly affect the translocation, residence and availability of contaminants in the soil; therefore, they are extremely important in terms of the environmental risk assessment of areas with excessive potential and current pollution deposition in soils.

## Figures and Tables

**Figure 1 molecules-28-00429-f001:**
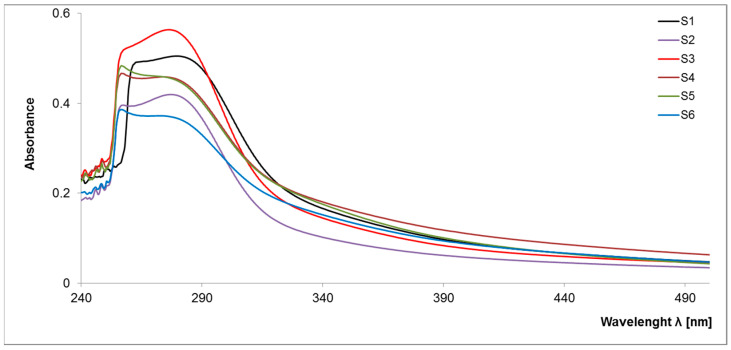
Spectral characteristics of the S-SOM in the UV–VIS range.

**Figure 2 molecules-28-00429-f002:**
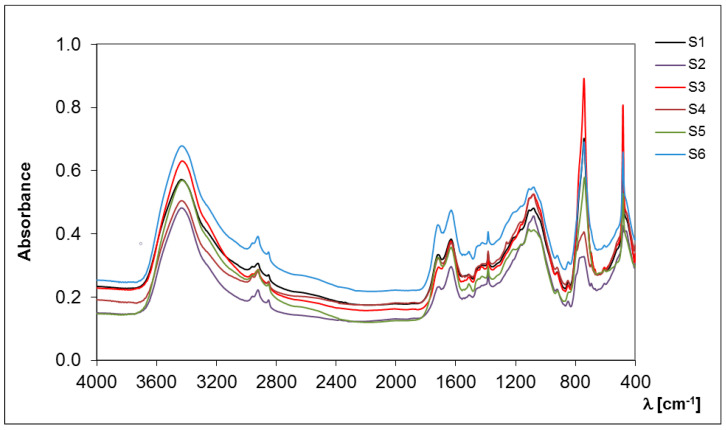
Spectral characteristics of the S-SOM in the infrared range.

**Figure 3 molecules-28-00429-f003:**
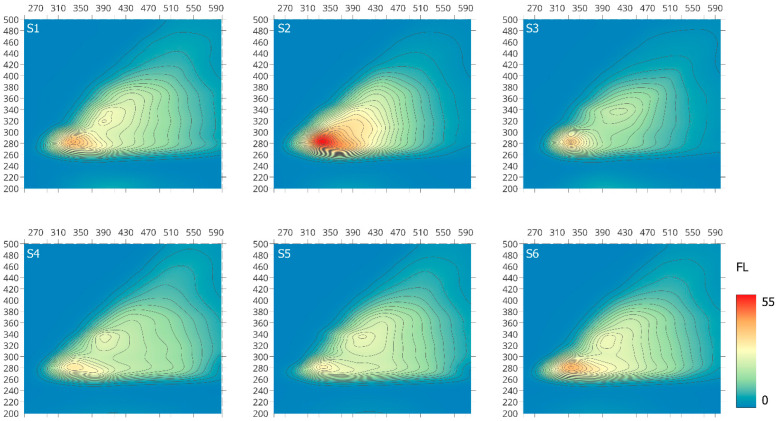
Three-dimensional EEM fluorescence spectra of the S-SOM.

**Figure 4 molecules-28-00429-f004:**
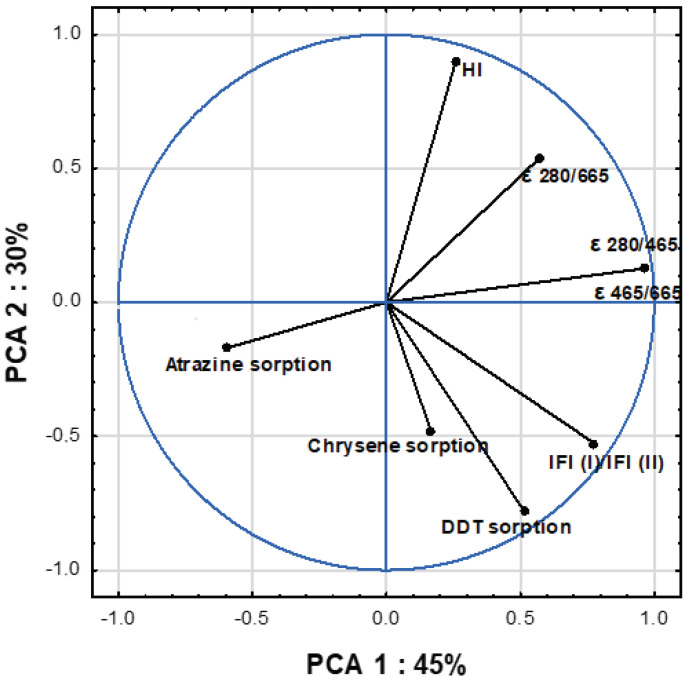
PCA-ordination biplot (component 1 and component 2) and eigenvectors of correlation matrix used to generate the PCA components.

**Table 1 molecules-28-00429-t001:** Molar absorption coefficients values for extracted S-SOM (n = 6).

S-SOM Sample	ε 280	ε 465	ε 665	ε 280/465	ε 280/665	ε 465/665
S1	498.5	55.9	22.5	8.92	22.16	2.48
S2	417.4	40.1	19.9	10.40	20.98	2.02
S3	558.8	52.2	26.3	10.70	21.27	1.99
S4	453.3	75.6	31.7	6.00	14.30	2.38
S5	449.9	54.7	17.7	8.23	25.45	3.09
S6	366.4	57.5	22.9	6.38	16.03	2.51

**Table 2 molecules-28-00429-t002:** Hydrophobic/hydrophilic properties of S-SOM. Intensity values are derived from 2950–2830 cm^−1^ (hydrophobic component) and 1770–1560 cm^−1^ (hydrophilic component) areas of absorption bands of the FT-IR spectra.

Sample of S-SOM	Hydrophobic (Intensity)	Hydrophilic (Intensity)	Hydrophobicity Index (HI)
S1	0.698	17.175	0.041
S2	1.103	13.359	0.083
S3	1.112	17.109	0.065
S4	1.027	18.005	0.057
S5	1.247	22.176	0.102
S6	0.669	18.583	0.050

**Table 3 molecules-28-00429-t003:** Fluorescence intensity of IFl (I) and IFl (II) peaks and their proportion in S-SOM fraction extracted from soils.

Sample of S-SOM	IFl (I) [a.u.]	IFl (II) [a.u.]	IFl (I)/IFl (II)
S1	39.5	26.8	1.47
S2	54.5	n.d	-
S3	35.5	19.9	1.78
S4	31.5	24.7	1.27
S5	29.6	25.1	1.17
S6	40.0	25.6	1.56

n.d: non detected.

**Table 4 molecules-28-00429-t004:** Sorption equations calculated for OCs as influenced by S-SOM (y(OCs) = b^−ax(SOM_E_)^).

Sample of S-SOM	Sorption Equation		Total Accumulated Amount	
	Atrazine			
S1	y = 0.192e^−0.028x^	R^2^ = 0.986	99.93%	*Aa*
S2	y = 0.074e^−0.021x^	R^2^ = 0.922	87.52%	*Ba*
S3	y = 0.067e^−0.016x^	R^2^ = 0.923	99.91%	*Aa*
S4	y = 0.099e^−0.019x^	R^2^ = 0.926	99.78%	*Aa*
S5	y = 0.140e^−0.027x^	R^2^ = 0.932	99.84%	*Aa*
S6	y = 0.264e^−0.028x^	R^2^ = 0.887	99.84%	*Aa*
	DDT			
S1	y = 10.02e^−0.015x^	R^2^ = 0.903	74.35%	*Bb*
S2	y = 9.03e^−0.007x^	R^2^ = 0.876	75.50%	*Ba*
S3	y = 9.26e^−0.001x^	R^2^ = 0.977	79.99%	*ABb*
S4	y = 9.15e^−0.003x^	R^2^ = 0.989	64.67%	*Cb*
S5	y = 9.50e^−0.007x^	R^2^ = 0.920	64.22%	*Cb*
S6	y = 10.82e^−0.012x^	R^2^ = 0.956	81.64%	*Ab*
	Chrysene			
S1	y = 23.38e^−0.002x^	R^2^ = 0.935	79.82%	*Ab*
S2	y = 17.94e^−0.006x^	R^2^ = 0.924	48.56%	*Cc*
S3	y = 14.89e^−0.006x^	R^2^ = 0.946	45.24%	*Cc*
S4	y = 21.05e^−0.007x^	R^2^ = 0.932	35.23%	*Dc*
S5	y = 11.19e^−0.001x^	R^2^ = 0.929	49.15%	*Cc*
S6	y = 14.85e^−0.009x^	R^2^ = 0.902	60.40%	*Bc*

Uppercase letters indicate significant differences between OCs sorption on different S-SOM samples, while lowercase letters indicate significant differences between OCs sorption on the S-SOM extracted from the same samples (ANOVA, U Mann–Whitney’s test, *p* < 0.05).

**Table 5 molecules-28-00429-t005:** Table of PCA factor loadings matrix.

Parameters	PCA 1	PCA 2	PCA 3
ε 280/465	0.96	0.13	−0.08
ε 280/665	0.67	0.54	−0.71
ε 465/665	0.96	0.13	−0.08
IFl (I)/IFl (II)	0.78	−0.53	0.33
HI	0.26	0.90	0.13
Atrazine sorption	−0.70	−0.17	−0.42
DDT sorption	0.71	−0.78	0.07
Chrysene sorption	0.17	−0.48	−0.77
% of variance	45	30	18
Cumulative %	45	75	93

Loading ≥ 0.5 is shown in bold.

**Table 6 molecules-28-00429-t006:** Soil physicochemical properties (n = 6).

Soil Properties	S1	S2	S3	S4	S5	S6
Clay (%)	1	1	1	2	0	8
Silt (%)	66	7	70	17	0	33
Sand (%)	33	92	29	81	0	59
WRB taxonomy	Fluvisols	Luviosols	Luviosols	Luviosols	Fluvisols	Leptosols
pH in KCl	7.3	6.4	6.9	7.1	7.3	7.4
TOC (g kg^−1^)	19.2	19.3	16.9	15.0	58.7	29.4
TN (g kg^−1^)	2.2	1.9	1.8	1.4	6.6	2.9
TC/TN	10.8	11.4	10.2	14.8	11.4	13.4

S1–S6: number of soil samples selected for the S-SOM extraction.

**Table 7 molecules-28-00429-t007:** Chemical and physical properties of the tested organic compounds used in the sorption experiment.

	Atrazine	Chrysene	44′DDT
Molecule structure	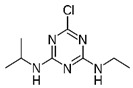	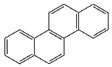	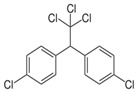
Type of compound	Pesticide (herbicide)	Polycyclic aromatic hydrocarbons	Pesticide (insecticide)
Molecular weight (g mol^−1^)	215.68	228.29	354.49
Water solubility (mg l^−1^)	35	0.0015	0.006
log K_o/w_	2.7	5.91	6.91
Vapor pressure (mPa)	0.039	1.04·10^−6^	0.025
Soil degradation, DT 50 (days)	75	120	6200
Bioconcentration factor (l kg^−1^)	4.3	0.00033	3173

Information downloaded from the Hazardous Substances Data Bank (HSDB) and the Pesticide Properties Database (PPD).

**Table 8 molecules-28-00429-t008:** GC-MS/MS method conditions.

Parameter	GC-MS/MS
Injection mode	Hot-splitless; MMI injection mode
Injection volume	2 μL
Inlet temperature	280 °C
Carrier gas	He, constant flow 1.00 mL min^−1^ (column 2 = 1.20 mL min^−1^)
Detector temperature	-
Makeup gas	-
Oven program	70 °C for 2 min 25 °C/min to 150 °C for 0 min; 3 °C/min to 200 °C for 0 min; 8 °C/min to 280 °C for 10 min hold time
MS transfer line temperature	280 °C
Backflush settings	5 min during post-run / 310 °C
Aux EPC pressure	~50 psi
Inlet pressure	~2 psi
Column pressure	~3 psi
Electron energy	70 eV
MS1 and MS2 resolution	Wide
Collision cell	1.5 mL min^−1^ N_2_ and 2.25 mL min^−1^ He
Source temperature	300 °C
Quad temperatures	150 °C

## Data Availability

The entire set of raw data is available in the resources of the authors of publication.

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
