# Peer review of "Sorption of Organic Contaminants by Stable Organic Matter Fraction in Soil"

_molecules, 2023, doi:10.3390/molecules28010429_

Round 1
Reviewer 1 Report
Dear authors,
the manuscript submitted is very interesting and delivers additional insights into the stuctural and functional nature of SOM and related interactions to organic pollutants.
The scientific structure of the work is well organized and the description of the various steps of the experimental protocol is appropriate.
Material and methods are accurately described. At the same time the experimental protocol should be revised in order to present more clearly the procedures, the preparation of the soil sample, the sampling steps, and so on.
This will enable the reader to immediately understand the overall approach of the experiments.
The results and discussion also deserve to be reconsidered. The variability of the 6 soil samples in terms of sorption/desorption properties, the structural model the authors think about for the S-SOM should be more stressed, better defined in order to result in an immediate comprehension of the effort produced.
The experimental protocol is very dense and several analytical and instrumental approaches are used and well presented. Improving the discussion and presenting some highlights will result in a very interesting paper.
The discussion related to the sorption phenomena needs also additional speculations which may improve also the understanding of the results. In particular, there is no sufficient description of the possible structural and conformational nature of the SOM fractions, which to a certain extent may strongly influence the environmental dispersion of the OCs.
In general, the manuscript needs a thorough revision of the language. Some sentences should be rephrased and few grammar corrections have to be made.
A short example is reported in the attached file.

Author Response
Reviewer #1
the manuscript submitted is very interesting and delivers additional insights into the stuctural and functional nature of SOM and related interactions to organic pollutants.
The scientific structure of the work is well organized and the description of the various steps of the experimental protocol is appropriate.
Material and methods are accurately described. At the same time the experimental protocol should be revised in order to present more clearly the procedures, the preparation of the soil sample, the sampling steps, and so on.
This will enable the reader to immediately understand the overall approach of the experiments.
Resp. The soil sampling procedure has been expanded and supplemented with missing information as follow ” The six soil samples (S1-S6) were collected from the surface layer (0-30 cm) of agricultural lands, including arable fields. The distribution of sampling points was aimed to reflect various soil properties, preliminarily based on maps (1:25,000 scale; database of the Institute of Soil Science and Plant Cultivation). For each sampling site six replicate samples were collected at the center and vertices of 1 m x 1 m square, averaged on the site after removal of the upper layer of organic vegetative materials, and mixed to provide a bulked sample for the site. The collected soil samples were air dried and passed through a 2 mm mesh sieve. All samples were then stored in the dark at 12-16oC before the determination of laboratory analysis.” (l. 429-438).
The results and discussion also deserve to be reconsidered. The variability of the 6 soil samples in terms of sorption/desorption properties, the structural model the authors think about for the S-SOM should be more stressed, better defined in order to result in an immediate comprehension of the effort produced.
Resp. Some sentences have been reworded/rephrased to the better for better highlighting of the obtained results especially section 2.3 (l. 337-341, 380-425, 609-612).
The experimental protocol is very dense and several analytical and instrumental approaches are used and well presented. Improving the discussion and presenting some highlights will result in a very interesting paper.
The discussion related to the sorption phenomena needs also additional speculations which may improve also the understanding of the results. In particular, there is no sufficient description of the possible structural and conformational nature of the SOM fractions, which to a certain extent may strongly influence the environmental dispersion of the OCs.
Resp. Key information on the structural structure and sorption potential of SOM structures has been added (l. 380-425).
In general, the manuscript needs a thorough revision of the language. Some sentences should be rephrased and few grammar corrections have to be made.
A short example is reported in the attached file.
Resp. Thank you very much for your effort in reviewing the manuscript. Your comments are very valuable and each one has been considered and applied in the revised paper. Moreover the paper has been corrected by the proofreading service and I hope it will meet with your approval.
Reviewer 2 Report
Another sentence should follow the first line, in order to clarify why accumulation of OCs in soil is an important problem.
The first time a phrase, such as OCP, NCP, PAH, etc, should appear in its full form.
Six different soil types had not been clearly defined.
Principal component analysis was used in this study; however, in the method, it was not mentioned.
Line 16 and 25: S-SOM, S-SOME should be uniformed,
Line 19: OCPs, NCPs should mention the full form as line 76
Line 21: Soil organic matter, it is abbreviation
Line 92, the first sentence is repeated in the abstract.
Line 415: Total organic carbon it is abbreviation
Line 426: Song et al (2014), it should change number
Author Response
Reviewer #2
Another sentence should follow the first line, in order to clarify why accumulation of OCs in soil is an important problem.
Resp.: For clarification, the first sentence has been reworded as follows: “Accumulation of toxic organic pollutants (OCs) in the soil is an important environmental problem due to their harmful properties and the risk of migration into living organisms creating a threat to their health and life.” (l. 13-15)
The first time a phrase, such as OCP, NCP, PAH, etc, should appear in its full form.
Resp.: All abbreviations have been accordingly explained in the text.
Six different soil types had not been clearly defined.
Resp.:The soil physicochemical properties have been included in table 6 and described according to the sentence “The research was conducted as a batch experiment. For this purpose a stable fraction of organic matter (S-SOME) was separated from six soils (characterized by diversified properties: TOC=15.0-58.7 g∙kg-1; TN=1.4-6.6 g∙kg-1, pH in KCl=6.4-7.4 and WRB taxonomy: Fluvisols, Luviosols, Lepto-sols)” in l. 452-456 .
Principal component analysis was used in this study; however, in the method, it was not mentioned.
Resp.: The appropriate description of the PCA method has been added to the section of 3.3 Statistical analysis: “Moreover, the Principal Component Analysis (PCA) were applied to assess the relation-ship between molecular sorption properties of isolated S-SOM and sorption affinity ex-pressed by maximum sorption amount of OCs”. (l. 609-612).
Line 16 and 25: S-SOM, S-SOME should be uniformed,
Resp.: The abbreviation S-SOM stands for 'stable organic matter' while S-SOME stands for stable organic matter isolated from soil by extracting 'extractable fraction of stable soil organic matter'. This has been detailed explain in the ‘Materials and methods’ section (l. 609-612) Nevertheless, the nomenclature was unified by introducing one S-SOM designation in whole text.
Line 19: OCPs, NCPs should mention the full form as line 76
Resp.: The abbreviations of OCPs, NCPs have been accordingly explained (l. 19-21).
Line 21: Soil organic matter, it is abbreviation
Resp.: The previously introduced abbreviation “SOM” was used instead of the full name for “soil organic matter”
Line 92, the first sentence is repeated in the abstract.
Resp.: The abstract is a short version of the publication and should reflect the contents of the manuscript as accurately as possible. In our opinion, such repetitions are not a mistake, but only correspond to the topic discussed in the publication.
Line 415: Total organic carbon it is abbreviation
Resp.: The designation has been accordingly implemented.
Line 426: Song et al (2014), it should change number
Resp.: The reference has been marked as number ‘54’ according to the literature list.
Round 2
Reviewer 1 Report
Dear authors,
I appreciate very much the effort in revising the manuscript entirely.
I believe the manuscript is very interesting and deserves to be published in the present state.